# Large-Area Biocompatible Random Laser for Wearable Applications

**DOI:** 10.3390/nano11071809

**Published:** 2021-07-12

**Authors:** Kun Ge, Dan Guo, Xiaojie Ma, Zhiyang Xu, Anwer Hayat, Songtao Li, Tianrui Zhai

**Affiliations:** 1Faculty of Science, College of Physics and Optoelectronics, Beijing University of Technology, Beijing 100124, China; GEKUN@emails.bjut.edu.cn (K.G.); dguo@bjut.edu.cn (D.G.); xiaojiema@emails.bjut.edu.cn (X.M.); xu.zhiyang@hotmail.com (Z.X.); anwerhayatnoor@gmail.com (A.H.); 2Department of Mathematics & Physics, North China Electric Power University, Baoding 071000, China; songtaoli2001@126.com

**Keywords:** random laser, biocompatible, large-area, polymer film, wearable

## Abstract

Recently, wearable sensor technology has drawn attention to many health-related appliances due to its varied existing optical, electrical, and mechanical applications. Similarly, we have designed a simple and cheap lift-off fabrication technique for the realization of large-area biocompatible random lasers to customize wearable sensors. A large-area random microcavity comprises a matrix element polymethyl methacrylate (PMMA) in which rhodamine B (RhB, which acts as a gain medium) and gold nanorods (Au NRs, which offer plasmonic feedback) are incorporated via a spin-coating technique. In regards to the respective random lasing device residing on a heterogenous film (area > 100 cm^2^), upon optical excitation, coherent random lasing with a narrow linewidth (~0.4 nm) at a low threshold (~23 μJ/cm^2^ per pulse) was successfully attained. Here, we maneuvered the mechanical flexibility of the device to modify the spacing between the feedback agents (Au NRs), which tuned the average wavelength from 612.6 to 624 nm under bending while being a recoverable process. Moreover, the flexible film can potentially be used on human skin such as the finger to serve as a motion and relative-humidity sensor. This work demonstrates a designable and simple method to fabricate a large-area biocompatible random laser for wearable sensing.

## 1. Introduction

Random lasers [1,2,3] have been extensively investigated for their unique optical properties and potential applications in speckle-free imaging [4,5,6] and sensors [7,8]. In particular, large-area and biocompatible random lasing for wearable sensors has attracted considerable attention [9,10,11]. Random lasing output works as a signal to monitor human activities and detect relative humidity. The wearable polymer film has the ability to work with partial damage. The biocapacity of the heterogeneous film is important for wearable sensors, which can potentially be used on human skin, such as the finger, to detect vital signs and monitor secreted sweat levels [12,13]. Random lasers based on localized surface plasmon resonance (SPR) of noble metallic nanoparticles have been reported in both thin films and solutions by researchers [14,15,16,17,18,19]. Hrelescu. et al. reported that hybrid multilayered plasmonic nanostars can reduce the pumping threshold for random lasing [20]. Follow-up demonstrations have motivated researchers to study localized surface plasmon resonance random multimode lasers in polymer thin films doped with gain dye and silver nanoparticles [21].

In addition, metallic nanoparticles (NPs) with periodic arrangement can act as a grating and lead to distributed feedback lasers by the effects of enhanced localized surface plasmon resonance and scattering [22,23,24,25]. The plasmonic resonance of metallic nanoparticles shows effects on random lasing output from visible to near-infrared ranges due to the enhancement of SPR and scattering in microcavities [26,27,28,29,30,31]. At the same time, the lasing emission efficiency and absorbance of gain materials are improved by the NPs. However, a very minor research contribution has been made with the allocation of the gold nanorods (Au NRs) on the surface (as a feedback agent) of large-area biocompatible heterogeneous film to realized wearable coherent random lasing. 

In this paper, a designable and simple method is used to fabricate a large-area biocompatible random laser for wearable applications by spin coating the polymer Rhodamine B (RhB) dissolved in dichloromethane on a glass substrate with Au NR_S_ on the surface. The polymethyl methacrylate (PMMA) film works as a matrix with RhB molecules distributed inside, while the Au NRs provide plasmonic feedback. Experimentally, the optical excitation of the heterogeneous film (area > 100 cm^2^) by a nanosecond laser with 532 nm wavelength can emit a coherent random lasing with a narrow linewidth (0.4 nm) at a low threshold of about 23 μJ/cm^2^ per pulse. The strong confinement mechanism provided by the active waveguide layer is the key for a narrow-band and low-threshold coherent random laser [32]. The range of tunable average wavelength is from 624 to 612.6 nm under mechanical bending and is recoverable. Furthermore, the transferred polymer film to the human skin and fingers can tune the coherent random emission wavelengths by mechanical stretching and relative humidity, which in turn can be operated as a wearable sensor to monitor human activities and detect sweating (relative humidity) from the body. The PMMA film does not harm human tissues and organs, showing biocompatibility for wearable applications.

## 2. Materials and Methods

The design and fabrication process of the large-area biocompatible random laser is illustrated in Figure 1a. The typical light-emitting molecule RhB (Tianjin Fuchen Chemical Reagents Factory, Tianjin, China) was used as the active material in the polymer film. The fabrication process of the polymer film is as follows: first, the Au NRs in the dichloromethane solution (Au NRs at 0.02 mg/mL) was spin-coated on the substrate at a speed of 1500 r/min for 30 s. The length and diameter of the Au NRs were about 50 and 25 nm, respectively, as shown in Figure 1d, thus the length to diameter ratio was 2:1. The Au NR_S_ were distributed on a silica slab to serve as scattering particles, provide coherent feedback in the large-area polymer film, and enhance the emission of RhB due to localized surface plasmon resonance in the local electric field. The RhB and PMMA were dispersed in the dichloromethane solvent with a concentration of 6 and 200 mg/mL, respectively. Then, they were mixed into a volume ratio of 1:1 under magnetic stirring for 30 min. Next, the mixture was spin-coated on the substrate at a speed of 1500 r/min for 30 s. The polymer film of PMMA and RhB was solidified after heating at 70 ℃ for 30 min.

After the solidification of the polymer film, a large-area biocompatible random laser was attained by peeling it from the substrate. The optical picture of the large-area biocompatible random laser is shown in Figure 1b. The large-area polymer film was flexible and can therefore be transferred to human skin and fingers, serving as a wearable sensor to detect vital signs and monitor relative humidity. The random laser does not harm human tissues and organs, thus it is biocompatible in wearable applications [9,10,11].

As shown in Figure 1c, the peak wavelengths of the normalized absorbance (red dashed line) and photoluminescence (red solid line) spectra of RhB were 552 and 601 nm, respectively. The extinction (black solid line) spectrum of Au NRs overlapped with the absorbance and photoluminescence (PL) spectra of the gain material. These can greatly improve lasing emission efficiency. The scanning electron microscopy (SEM) image of the Au NRs (50 nm) is illustrated in Figure 1d. 

The electric field intensity distribution in the transverse cross-section was numerically simulated with the commercial software COMSOL multi-physics 5.4. Typical localized electric field distributions demonstrated that transversal surface plasmon resonance (TSPR) mode is at 520 nm (in Figure 1e) and longitudinal surface plasmon resonance (LSPR) mode is at 650 nm (in Figure 1f). The simulation results demonstrated that the TSPR can enhance the pump, and LSPR can enrich the enhanced emission. The dispersed Au NRs on the surface of a large-area polymer film can serve as scattering particles to provide coherent feedback in the large-area polymer film, as well as enhance the emission of RhB due to SPR. The excellent overlap between the plasmonic resonance spectrum of the Au NRs and the PL spectrum of the active material (in Figure 1c) helped to lower the threshold of a random laser by SPR enhanced fluorescence. The Au NRs in the polymer film can serve as a coherent localized cavity to narrow the line-width of the random laser spectrum in the random system. Similarly, we can say that a strong confinement mechanism is essential for the narrow-band and low-threshold operation of the random laser.

## 3. Results and Discussion

A nanosecond laser with a wavelength of 532 nm (second harmonics from a 1064 nm Yb: YAG laser, with a repetition frequency of 10 Hz, and pulse width of 1 ns) was used as the pump source to investigate the output of lasing. The emission spectra of coherent random lasing were demonstrated at different pump power densities, as shown in Figure 2a. As the pump fluence of the optical excitation source increased from 10 to 90 μJ/cm^2^ per pulse, the emission spectrum from the prepared device indicated an evident narrow and protruded superlinear increase in intensity, which provided a confirmed analysis regarding the coherent random lasing action, as illustrated in Figure 2a. Only a broad spontaneous emission spectrum peak at 626 nm was observed when the pump fluence was lower than 23 μJ/cm^2^ per pulse. Hence, several discrete narrow peaks were observed when the power density exceeded 23 μJ/cm^2^ per pulse, indicating that the coherent resonant feedback had accumulated in the polymer film. The emission spectra were recorded using a spectrometer by Ocean Optics model Maya Pro 2000 with a spectral resolution of 0.1 nm. The magnified plotted curve of the pump fluence at 23 μJ/cm^2^ per pulse had a linewidth of approximately 0.4 nm, indicating the evolution point of random lasing, as depicted in Figure 2a, top right inset. The quality factor was over 1500. Figure 2b presents the output lasing intensity and full width at half maximum (FWHM) of the large-area polymer film through pumping at different positions with a distance step of 1 cm. The FWHM of coherent random lasing had almost no change at about 0.4 nm, demonstrating that the large-area coherent random lasing had excellent optical stability (see Appendix A). The position of the optical excitation source started at zero and increased in increments of 1cm, covering a total distance of 10 cm, as shown in inset of Figure 2b.

This was the entire intensity integrated over all possible emission modes, and the maximum intensity of random laser was chosen per Figure 2c. Figure 2c presents the evolution of the random lasing intensity (blue circles) and FWHM (red triangles) as a function of the pump fluences. The output intensity and FWHM were measured at different pump power densities with a lasing threshold of 23 μJ/cm^2^ per pulse, as shown in Figure 2c. The output curve exhibits a typical “S” shape, which is a clear indication of the transition from a spontaneous emission to an amplified spontaneous emission, and then to a stimulated emission with increasing power densities. This significant feature denotes that the coherent random laser had a low-working threshold, indicated by the black arrow in Figure 2c. The left illustration indicates the optical image under the pumping; the diameter of the pump spot was about 0.28 cm. The right inset shows partially enlarged FWHM values and pump fluence above the threshold, indicating the FWHM of lasing mode was about 0.4 nm. The PL spectra of PMMA film without Au NRs is incoherent random laser (see Appendix A).

To calculate the effective cavity length of the coherent random laser, the power Fourier transforms (PFT) of the spectrum were calculated and presented in Figure 2d. 

The spatial dimensions were calculated by
(1)$$p_{m}=\frac{mnL_{c}}{\pi }$$
where *p_m_* is a Fourier component, *m* is the order of the Fourier harmonics, *n* is the refraction index of the gain medium, and *L_c_* is the localized cavity dimension [33,34]. We consistently chose the first Fourier component as the numerical calculation, due to the resemblance to the actual situation. In our experiment, *m* and *n* were equal to 1 and 1.49, respectively, and *p*_1_ was equal to 17.14 μm, as shown in Figure 2d. Therefore, the effective optical cavity length *L_c_* was calculated to equal 36.14 μm by Equation (1).

The random laser was lifted away from the glass substrate and showed a bendable and flexible property. It was easily deformed under mechanical bending with the potential to be transferred to human skin. There were two steps in realizing the wearable applications while using the proposed random laser. First, the optical excitation should be replaced with indirect pumping or electrical pumping [35]. Regarding indirect pumping, the laser diode (LD) or light-emitting diode (LED) can be used as a pump source. Second, the emission spectrum of the random laser can be guided by optical fibers and collected by micro-detection when the sensor is worn.

Based on this design, the biocompatible random laser can be applied on the shoulder, elbow, and palm for wearable sensors. In the experiment, the polymer film was bent by two translation stages to imitate a human body’s bending motion, as illustrated in Figure 3a. The schematic diagram of the principle of bending strain is shown in Figure 3b. Additionally, the average wavelength of the coherent random laser was blue-shifted due to the decrease of the distances between Au NRs when the polymer film was in bending strain, which altered the plasmon interaction and scattering. Here, the length (L = 1 cm) was defined as the original length of the polymer film without exerting any bending strain and ΔL was the bending length when the polymer film was under bending strain. Figure 3c,d shows the evolution of the PL spectra of the RhB polymer film ornament with Au NRs under different bending lengths from 0 to 5 mm and was recoverable, which exhibited good repeatability. The pump fluence was about 80.26 μJ/cm^2^ per pulse.

Additionally, the wavelength of the coherent random laser was blue-shifted due to the decrease of the distances between Au NRs, which altered the plasmon interaction and scattering. In our experiment, the maximum blue-shifting over 11 nm (average wavelength ranges from 612.6 to 624 nm) was achieved with a bending length of 5 mm. The experimental data of the average wavelength are shown in Figure 3e, and agree with the observed blue-shifted amount of the random lasing emissions shown in Figure 3c,d. The spectra are especially different from others due to the spectra and were recorded at different times and the different positions.

Here, we investigated the lasing recoverability of the polymer film. Two sides of the polymer film were fixed into the clips. The translation stage can tune the curvature of a polymer film. Under different bending lengths from 0 to 5 mm, the average wavelength was blue-shifting from 624 to 612.6 nm. The lasing mode returned to the original wavelength when the polymer film regained its original shape (length), as shown in Figure 3c,d, which illustrates that the random lasing sensor had excellent recoverability. The blue-shifting of the average wavelength relies on the bending degree (in Figure 3e). The blue-shifting was more than 11 nm (average wavelength ranged from 624 to 612.6 nm). The plotted data points were estimated from the experimental results of Figure 3c. The random laser film can be transferred to the human wrist. The results prove that the large-rea biocompatible random laser can serve as a wearable sensor to track tissue bending.

One of the features of random lasing is that the polymer film can be transferred to human tissue, such as the fingers, wrist, neck, arms, etc (see Appendix A). The human body secretes sweat during physical exercise. The polymer film is sensitive to relative humidity (RH) so that the random lasing polymer film can serve as an RH sensor. Figure 4a presents the schematic illustration of the wearable sensor attached to different body parts. When the RH of the environment increases, the average refractive index of the polymer film decreases, owing to absorbing the water molecules in the air and leading to the blue-shifting of the spectrum of the random laser [36]. In addition, the concentration of gain material decreases with water molecules adsorbed into the polymer film, which can also cause the blue-shifting of the lasing mode. Figure 4b presents the schematic diagram of the principle of RH sensing.

During the measurement, the wearable RH sensing setup was based on a humidity control system. The emission from biocompatible random lasing was collected by the same optical fiber. The increased concentration of the RH from 40% to 88% introduced a blue shift in the emission peaks of the polymer film, as depicted in the Figure 4c. To investigate the effect of the wavelength shift of the random lasing due to the RH, we observed a significant blue-shift of the average wavelength of random laser with an increase the relative humidity (see Appendix A). Figure 4c presents the relationship between wavelength shift and RH, in which the RH range was from 40% to 88%, with a slope of approximately 0.1 nm/%. All experimental data derives from Figure 4c.

The limit of detection (LOD) for realistic humidity sensing was calculated from LOD = 3.3σ_D_/b [37,38], where b is the slope of the FWHM value vs. the realistic humidity curve and σ_D_ is the standard FWHM deviation. Then, the emission peaks were tracked continuously over 300 s at a 40% RH, during which the peak positions remained relatively stable with a wavelength shift fluctuation of about 0.07 nm standard deviation. Thus, the calculated LOD of the RH sensing was 2.31% RH, which is moderately sensitive.

Thus, the large-area random laser can serve as a wearable sensor to detect human vital signs when the polymer film is transferred to human skin. Random lasing shows no harm to human tissues and organs, indicating that it has biocompatibility for wearable applications.

## 4. Conclusions

In summary, we experimentally fabricated a large-area biocompatible random laser by lift-off technique for wearable sensors. Upon optical excitation, the evolution of coherent random lasing from the polymer film was realized at a low threshold of 23 μJ/cm^2^ per pulse, which comprises a linewidth of about 0.4 nm. The lasing spectrum can be tuned by mechanical bending and relative humidity. Therefore, our work provides a route toward wearable sensors, where the transferred polymer film to human skin can be used to detect vital signs and relative humidity. Large-area random lasing shows no harm to human tissues and organs, and biocompatibility for wearable applications. This work demonstrates a designable and simple method to fabricate large-area biocompatible random lasers for wearable sensors.

## Figures and Tables

**Figure 1 nanomaterials-11-01809-f001:**
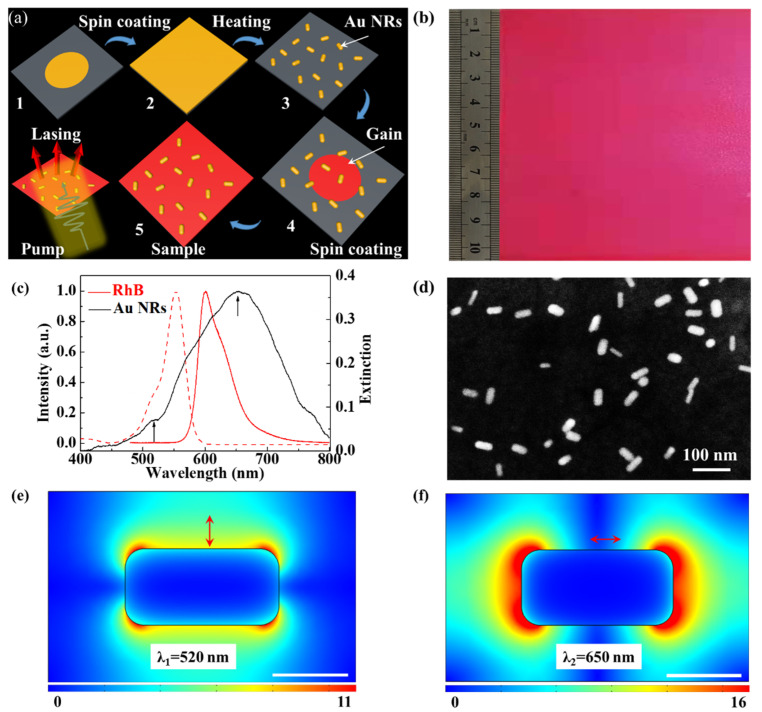
Schematic and imaging of the sample. (**a**) The design and fabrication process of a large-area biocompatible random laser for wearable sensors and their surface with Au NR_S_ is illustrated. (**b**) The optical image is the large-area biocompatible polymer film, the area of which was over 100 cm^2^. (**c**) Normalized absorbance (red dashed line), photoluminescence (red solid line) spectra of RhB, and the extinction (black solid line) of Au NRs. The arrows indicate the plasmonic resonant peak of the Au NRs. (**d**) The scanning electron microscopy image of the Au NRs (scale bar = 100 nm). (**e**) Numerically simulated electric field distribution of TSPR mode at 520 nm. (**f**) Numerically simulated electric field distribution of LSPR mode at 650 nm. Scale bar = 50 nm.

**Figure 2 nanomaterials-11-01809-f002:**
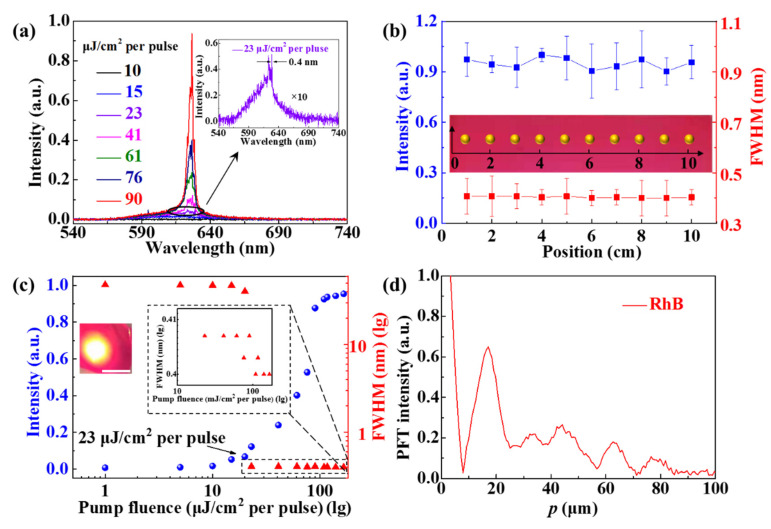
Spectra characterization of the coherent random laser. (**a**) The evolution of random lasing under different pumping fluences ranged from 10 to 90 μJ/cm^2^ per pulse. Inset: the enlarged view of the emission spectrum with 23 μJ/cm^2^ per pulse. (**b**) The FWHM (red squares) and the output intensity (blue squares) were at different pump positions. (**c**) The pump power and FWHM were at different power densities with the lasing threshold of 23 μJ/cm^2^ per pulse. Scale bar is 0.3 cm. (**d**). The power Fourier transform of the random laser spectra.

**Figure 3 nanomaterials-11-01809-f003:**
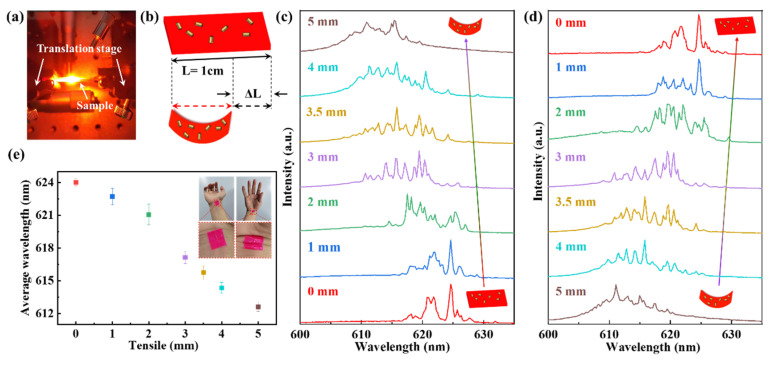
The optical spectrum characteristics were under the bending strain. (**a**) A schematic diagram of the experimental device was under the pumping light. The polymer film was fixed into the clip, and the translation stage can tune the curvature of a polymer film. (**b**) The schematic diagram of the principle of bending strain. The original length (L) of the polymer film was 1 cm, ΔL is the bending length when the polymer film is under bending strain. (**c**,**d**) The signals of the coherent random laser were detected under different bending strains. Insets exhibit schematic diagrams of a polymer film under bending and regaining state. (**e**) The blue-shifting of the average wavelength is a function of the degree of bending. Top right inset: two states of a polymer film transferred to the wrist.

**Figure 4 nanomaterials-11-01809-f004:**
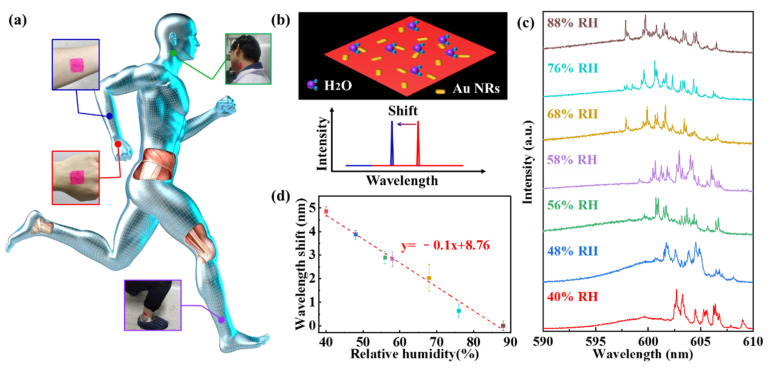
Measurement of small motion signals with polymer film sensors. (**a**) The wearable sensors for on-skin applications. The schematic illustration of the sensor attached to different parts for detecting subtle human motions. (**b**) The schematic diagram of the principle of RH sensing. (**c**) The emission spectra of the random laser under different relative humidities ranging from 40% to 88%. (**d**) Plots describing blue-shifting emission peaks as a function of relative humidity.

## Data Availability

Data are publicly available and cited in accordance with journal guidelines.

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
