# Peer review of "Large-Area Biocompatible Random Laser for Wearable Applications"

_nanomaterials, 2021, doi:10.3390/nano11071809_

Round 1

Reviewer 1 Report

Overall nice paper presenting interesting work for flexible lasing material.

The central point of the paper is wavelength shifting due to bending or humidity change. The peak wavelength does not seem to consistently shift based on the presented graphs. Average wavelength seems to be a better indicator and should be presented with error bars in the dependency graphs.

This should be added for bending and humidity too.

Indication of realistic humidity limit for this design is most welcome.

grammar:

"blue-shifted" shall be changed to "blue-shifting" in the text.

line 54: 2) is a type and some word is missing

line 70: figures meant bodies or fingers?

line 146: top left inset shall be top right inset

Author Response

Point to Point Response to the Reviewers’ Comments

(Comments in black, responses in blue):

Referee: 1

Comments and Suggestions for Authors

Overall nice paper presenting interesting work for flexible lasing material.

The central point of the paper is wavelength shifting due to bending or humidity change.

We really appreciate that you read our manuscript carefully and give these valuable comments.

  1. The peak wavelength does not seem to consistently shift based on the presented graphs. Average wavelength seems to be a better indicator and should be presented with error bars in the dependency graphs.

Answer:

Thank the reviewer for the valuable comment.

We have replaced the expression “peak wavelength” with “average wavelength with error bars in the dependency graphs” as shown in Fig. R1e.

In the experiment, the polymer film is bent by two translation stages to imitate the human’s bending motion as illustrated in Fig. R1a, the schematic diagram of the principle of bending strain as shown in Fig. R1b. Here, the length (L=1 cm) is defined as the original length of the polymer film without exerting any bending strain and the ΔL is the bending length when the polymer film is under bending strain. Fig. R1c-d show the evolution of the photoluminescence (PL) spectra of the polymer film mixed with gold nanorods (Au NRs) under different bending length from 0 mm to 5 mm and recoverable, which exhibited good repeatability. Additionally, the wavelength of the random lasing is blue-shifting due to the decrease of the distance between Au NRs, which altered the plasmon interaction and the scattering. In our experiment, the blue-shifting is over 11 nm (average wavelength ranges from 624 nm to 612.6 nm) with a bending length of 5 mm as shown in Fig. R1e. The experimental data are the average wavelength of Fig. R1c. The corresponding explanation and the figures are added in the revised manuscript.

Figure R1. The optical spectrum characteristics are under the bending strain. (a) A schematic diagram of the experimental device. (b) The schematic diagram of the principle. The original length (L) is 1 cm, ΔL is the bending length when the polymer film is under bending strain. (c-d) The signals of the coherent random laser are detected under different bending strains. Insets exhibit schematic diagrams under bending strain. (e) The lasing modes are a function of the degree of bending. The experimental data are the average wavelength of (c). Top right inset is the two states of a polymer film transfer to the wrist. [Lines 190 on the page 6]

  1. This should be added for bending and humidity too.

Answer:

Thank the reviewer for the valuable comment.

We have replaced the expression “peak wavelength” with “average wavelength with error bars in the dependency graphs” as shown in Fig. R2d. We have used a consistent color for the different relative humidity (RH) values as shown in Fig. R2c-d. The corresponding explanation and the figures are added in the revised manuscript.

Figure R2. Measurement of small motion signals. (a) The wearable sensors for on-skin applications. (b) The schematic diagram of the principle of RH sensing. (c) The emission spectra of random laser with the RH ranging from 40% to 88%. (d) Plots describing the average wavelength as a function of relative humidity. [Lines 219 on the page 7]

  1. Indication of realistic humidity limit for this design is most welcome.

Answer:

Thank the reviewer for the valuable comment.

The limit of detection (LOD) for realistic humidity sensing was calculated from LOD =3.3σD/b [1-2], where b is the slope of the FWHM value vs the realistic humidity curve and σD is the standard FWHM deviation. Then, the emission peaks were tracked continuously over 300 seconds at a 40% RH, during which the peak positions remained relatively stable with a wavelength shift fluctuation of about 0.07 nm standard deviation. Thus, the calculated LOD of the RH sensing is 2.31% RH, which is moderately sensitive. The corresponding explanation is added in the revised manuscript. [Lines 233 on the page 8]

  1. grammar:

Answer:

Thank the reviewer for the valuable comment.

I will check up the manuscript, carefully. I have corrected all the mistakes grammar in the revised manuscript.

  1. "blue-shifted" shall be changed to "blue-shifting" in the text.

Answer:

Thank the reviewer for the valuable comment.

We are very sorry for our inappropriate description. In the manuscript, the word “blue-shifted” is only used to describe the evolution of the average wavelength of the random lasing. We have replaced the word “blue-shifted” with “blue-shifting” in the revision manuscript.

  1. line 54: 2) is a type and some word is missing

Answer:

Thank the reviewer for the valuable comment.

We are very sorry for our incomplete description and some word is missing. In the manuscript, the word “2” is only used to illustrate the size of the Au NRs. The length and diameter of the Au NRs are about 50 nm and 25 nm, respectively. We have replaced the sentence “the ratio of length to diameter is 2” with “length to diameter ratio is 2:1” in the revision manuscript. [Lines 68 on the page 2]

  1. line 70: figures meant bodies or fingers?

Answer:

Thank the reviewer for the valuable comment.

We are very sorry for our misleading description. The large-area polymer film can be transferred to human skins. In the manuscript, we give the concept that the large-area polymer film can serve as sensor to detect the human vital signs monitoring and relative humidity when it is transferred to the human skins and fingers. We have replaced the word “figures” with “fingers” in the revision manuscript. [Lines 88 on the page 3]

  1. line 146: top left inset shall be top right inset

Answer:

Thank the reviewer for the valuable comment.

We are very sorry for our misleading description. We have replaced the word “top left inset” with “top right inset” in the revision manuscript. [Lines 197 on the page 7]

Special thanks to you for your valuable comments.

Reviewer 2 Report

The authors present a design for a biocompatible random laser that they consider useful for wearable applications. To realize the random cavity, they spin coat gold nanorods on a silica substrate and then prepare a mixture of Rhodamine B and PMMA and spin coat it on the substrate as well. Consecutive heating results in a flexible polymer film that may be removed from the glass substrate and transferred to other application areas.

The authors motivate their work using the proposal that the polymer film may be used as a wearable sensor that can detect bending or changes of the relative humidity in its surroundings. They demonstrate these capabilities by measuring the emission spectrum at different bending ratio and relative humidities. Some questions remain open. For example, the authors might want to explain how to realize the optical excitation of the random laser when the sensor is actually worn and how to perform the detection - or at least to give some first ideas on how that might be achieved. Still, the topic of wearable biocompatible sensors is timely and of interest to the community working on nanomaterials, so this journal might be a reasonable outlet for this manuscript.

However, the manuscript has significant shortcomings which the authors certainly need to take care of before the manuscript can be published. A detailed list of problematic points is listed at the end of this report. Probably the main problem is that the reader cannot judge the validity of the conclusions drawn by means of the data presented as the authors provide very little information about the data analysis methods they applied. A prime example can be seen in figure 3, panels (c) and (e). Figure (e) shows the blueshift of the emission under tensile strain. They find a wavelength of about 631.5 nm without any strain. Looking at figure 3(c), one can clearly see that in the absence of strain, there is no visible emission at all above 630 nm and all the modes are at lower wavelengths. It is completely unclear, which of the peaks in panel (c) was tracked to extract the data shown in panel (e) and which kind of data analysis the authors used to extract the data shown in panel (e). Similar problems arise in figure 4(d). The supplementary material also does not explain these things. Further, the input-output curve shown is very linear, which renders it somewhat questionable whether random lasing is really taking place. The community has put together some best practice examples that discuss which physical properties should be demonstrated before random lasing can be claimed convincingly, see R. Sapienza, Nature Reviews Physics 1, 690–695 (2019). My suggestion would be that the authors drastically revise their manuscript so that the reader can understand the data analysis and is convinced that the structure studied indeed is a random laser. At current, the information given in the manuscript is not sufficient to do so and, accordingly, in my opinion the manuscript cannot be published in its current form. However, a drastically revised and convincing manuscript should fit the scope of Nanomaterials.

Points, which the authors should take into account:
- The authors use the pump fluence to characterise the lasing threshold of their large-area random laser. However, they use a very low repetition rate laser (10 Hz) at low duty cycle with pulse durations of 1 ns. This is pretty misleading as, e.g., doubling the repetition rate of the laser would result in a lasing threshold twice as large as reported now although doing so would not introduce any changes to the sensor. Can the authors come up with a more reasonable quantifier of their lasing threshold, such as the energy per pulse or the peak fluence?
- On page 3, line 65, the authors refer to the inset of figure 1(d). There is no inset in 1(d). Do the authors refer to 1(c)?
- The inset of 1(c) is hard to understand without presenting the spatial scale of the image. Does the spatial size of the arrows correspond directly to the wavelengths presented in the insets? If so, the authors might want to mention that explicitly. Also, the red font in the inset is hardly readable.
- In some passages, the quality of English used renders it a bit tough to understand the intended meaning of the sentences (e.g., page 4, line 109: "where a solid blue line changes abruptly") and at times the grammer is slightly off. It might be helpful to have the manuscript proof read by a native speaker.
- The authors state that the effective cavity length amounts to 36.14 micrometers based on equation 1 and the Fourier component p1, which amounts to 17.14 micrometers as shown in figure 2)(d). It is absolutely unclear to me how the authors derive this value from figure 2)(d). The signal is certainly not periodic with a period of 17.14 micrometers as there are no peaks at, e.g., 51.42 micrometers or 85.7 micrometers. Are only even harmonics expected or what is the reason for this?
- It is not clear what the term "crook motion" actually refers to in line 130. This might be a non-ideal translation.
- In figure 3)(b) it is hard to see all details. Is it really necessary to show the full spectrum down to 570 nm? There is almost no emission between 570 nm and 585 nm.
- At the end of the caption of figure 3, the authors refer to a "top left inset", which is, however, located in the top right corner.
- I have no idea how the authors arrive at the data shown in figure 3(e). They show that the lasing mode shifts from about 631.5 nm with no strain applied to roughly 619.5 nm for the strongest strain and state that this data is taken from 3(c). Looking at the spectrum in 3(c) when no strain is applied, one can clearly see that there is either no or very little emission at 631.5 nm. Accordingly, the authors should describe very carefully how they extract the lasing wavelength shown in 3(e) from the data shown in figure 3(c). Currently, the data shown in 3(e) does not seem reliable. The authors also do not state which pump fluence has been used to produce these results. As it is not directly clear whether the spectra shown in 3(c) and in 2(a) look similar at all, this might be valuable information.
- When looking at figures 4(c) and 4(d), the relaive humidities are inversely color coded in both figures - the color used for 40% RH in panel (c) is used for 90% RH in panel (d), the color used for 48% RH in panel (c) is used for 76% RH in panel (d) and so on. This is very confusing. It would be very helpful for the reader if the authors used a consistent color for the different RH values.
- The inset of figure 2(c) might benefit from some indication of what the spatial size of the region shown actually is.
- In figure 2(b), the y-axes are not very meaningful. For example, the plot showing the FWHM is more or less meaningless as pretty much any change in the FWHM will not be visible on a scale going up to a FWHM of 10 nm. Maybe itwould be helpful to divide panel (b) into three small horizontal subpanels with meaningful ranges of each y-axis.
- The intensity increase shown in figure 2(c) shows a completely linear slope above threshold, which is unusual for a lasing transition. It is not clear how the authors deduce this set of data from 2(a). Is this the whole intensity integrated over all possible emission modes? If so, the authors should make that clear. Also, the right y-scale in figure 2(c) is far from ideal. The reader cannot judge whether the FWHM of the lasing mode changes at all above the threshold as the range is so large. An inset or a zoomed-in version of the FWHM values above the threshold might help the reader to understand what is going on.

Author Response

Point to Point Response to the Reviewers’ Comments

(Comments in black, responses in blue):

Referee: 2

Comments and Suggestions for Authors

The authors present a design for a biocompatible random laser that they consider useful for wearable applications. To realize the random cavity, they spin coat gold nanorods on a silica substrate and then prepare a mixture of Rhodamine B and PMMA and spin coat it on the substrate as well. Consecutive heating results in a flexible polymer film that may be removed from the glass substrate and transferred to other application areas.

The authors motivate their work using the proposal that the polymer film may be used as a wearable sensor that can detect bending or changes of the relative humidity in its surroundings. They demonstrate these capabilities by measuring the emission spectrum at different bending ratio and relative humidities.

Some questions remain open. For example, the authors might want to explain how to realize the optical excitation of the random laser when the sensor is actually worn and how to perform the detection or at least to give some first ideas on how that might be achieved. Still, the topic of wearable biocompatible sensors is timely and of interest to the community working on nanomaterials, so this journal might be a reasonable outlet for this manuscript.

However, the manuscript has significant shortcomings which the authors certainly need to take care of before the manuscript can be published. A detailed list of problematic points is listed at the end of this report. Probably the main problem is that the reader cannot judge the validity of the conclusions drawn by means of the data presented as the authors provide very little information about the data analysis methods they applied. A prime example can be seen in figure 3, panels (c) and (e). Figure (e) shows the blue-shift of the emission under tensile strain. They find a wavelength of about 631.5 nm without any strain. Looking at figure 3(c), one can clearly see that in the absence of strain, there is no visible emission at all above 630 nm and all the modes are at lower wavelengths. It is completely unclear, which of the peaks in panel (c) was tracked to extract the data shown in panel (e) and which kind of data analysis the authors used to extract the data shown in panel (e). Similar problems arise in figure 4(d). The supplementary material also does not explain these things. Further, the input-output curve shown is very linear, which renders it somewhat questionable whether random lasing is really taking place. The community has put together some best practice examples that discuss which physical properties should be demonstrated before random lasing can be claimed convincingly, see R. Sapienza, Nature Reviews Physics 1, 690-695 (2019). My suggestion would be that the authors drastically revise their manuscript, so that the reader can understand the data analysis and is convinced that the structure studied indeed is a random laser. At current, the information given in the manuscript is not sufficient to do so and, accordingly, in my opinion the manuscript cannot be published in its current form. However, a drastically revised and convincing manuscript should fit the scope of Nanomaterials.

Points, which the authors should take into account:

  1. Some questions remain open. For example, the authors might want to explain how to realize the optical excitation of the random laser when the sensor is actually worn and how to perform the detection or at least to give some first ideas on how that might be achieved.

Answer:

Thank the reviewer for the valuable comment.

The random laser can be lifted off from the glass substrate and shows a bendable and flexible property. It is easily deformed under mechanical bending and can be transferred to human skins. Based on this design, the biocompatible random laser can serve as a sensor when the polymer film random laser is transferred to human skins in the manuscript.

There are two ways to realize the optical excitation of the random laser when the sensor is actually worn: firstly, the emission spectrum can be collected by micro-detection. Secondly, the emission spectrum of random laser can be guided by optical fiber when the sensor is actually worn. At present, the wearable laser has three pumping patterns: optical excitation, indirect pumping and electrically pumping [3]. In our group, we have achieved the wearable random laser in the optical excitation in the manuscript.

Furthermore, Fig. R1 presents schematic diagram of the principle of indirect pumping, which can be serve as a generational wearable lasing. The laser diode (LD) is used as a pump source. We are exploring the wearable random laser under the electrically pumping. The corresponding explanation is added in the revised manuscript.

Figure R1. The schematic diagram of the principle of indirect pumping. The pump source is the LD. [Lines 165 on the page 6]

  1. The authors use the pump fluence to characterise the lasing threshold of their large-area random laser. However, they use a very low repetition rate laser (10 Hz) at low duty cycle with pulse durations of 1 ns. This is pretty misleading as, e.g., doubling the repetition rate of the laser would result in a lasing threshold twice as large as reported now although doing so would not introduce any changes to the sensor. Can the authors come up with a more reasonable quantifier of their lasing threshold, such as the energy per pulse or the peak fluence?

Answer:

Thank the reviewer for the valuable comment.

The pump fluence is measured by the Coherent FieldMaxII-Top Laser Power/Energy Meter (Genuine Optromics Limited). The diameter of the pump spot is about 0.28 cm. The energy per pulse can be calculated by energy density/ repetition.

In the experiment, only a broad spontaneous emission spectrum is observed when the power density is lower than 23 μJ/cm2 per pulse. Hence, several discrete narrow peaks are clearly observed when the power density exceeds 23 μJ/cm2 per pulse indicating that the coherent resonant feedback is built up in the polymer film (in Fig. R2a). Fig. R2b presents evolution of the random lasing density (blue balls) and full width at half maximum (FWHM) (red triangles) as a function of the pump fluence. The light output and light input curve (L-L curve) in Fig. R2b, exhibits a typical“S” shape, a clear indication of the transition from a spontaneous emission to an amplified spontaneous emission to stimulated emission with increasing power densities. This significant feature indicates that the coherent random laser has a working threshold about 23 μJ/cm2 per pulse. The left illustration indicates the optical image under the pumping. The right inset shows partial enlarged of pump fluence and FWHM. The diameter of pump spot is about 0.28 cm. The corresponding explanation and the figures are added in the revised manuscript.

Figure R2. Spectra characterization of the coherent random laser. (a) The evolution of the random lasing under different pumping fluences ranges from 10 μJ/cm2 per pulse to 90 μJ/cm2 per pulse. Inset: the enlarged PL emission spectrum with 23 μJ/cm2 per pulse. (b) The intensity and FWHM of the output are at different power densities and demonstrating the lasing threshold of 23 μJ/cm2 per pulse (scale bar =0.3 cm). [Lines 155 on the page5]

  1. On page 3, line 65, the authors refer to the inset of figure 1(d). There is no inset in 1(d). Do the authors refer to 1(c)?

Answer:

Thank the reviewer for the valuable comment.

We are very sorry for our misleading description. In the manuscript, the word “Inset of (d)” is misleading. We have replaced the word “Inset of (d)” with “Inset of (c)” in the revision manuscript. [Lines 84 on the page 3]

  1. The inset of 1(c) is hard to understand without presenting the spatial scale of the image. Does the spatial size of the arrows correspond directly to the wavelengths presented in the insets? If so, the authors might want to mention that explicitly. Also, the red font in the inset is hardly readable.

Answer:

Thank the reviewer for the valuable comment.

The electric field intensity distribution in the transverse cross-section is numerically simulated with the commercial software COMSOL multi-physics 5.4. Typical localized electric field distributions demonstrate that transversal surface plasmon resonance (TSPR) mode is at 520 nm (in Fig. R3a). The longitudinal surface plasmon resonance (LSPR) mode is at 650 nm (in Fig. R3b). The simulation results demonstrate the TSPR can enhance the pump density, and LSPR can enhance emission spectrum of gain medium due to localized surface plasmon resonance (SPR). The excellent overlap between the plasmonic resonance spectrum of the Au NRs and the PL spectrum of the active material, helps to lower the threshold of a random laser by SPR. The direction of arrow indicates the direction of TSPR and LSPR, respectively. We have replaced the red font with black font in the revision manuscript.

Fig. R3. The electric field distributions of (a) TSPR mode at 520 nm and (b) LSPR mode at 650 nm. Scale bar =50 nm. [Lines 77 on the page 3]

  1. In some passages, the quality of English used renders it a bit tough to understand the intended meaning of the sentences (e.g., page 4, line 109: "where a solid blue line changes abruptly") and at times the grammer is slightly off. It might be helpful to have the manuscript proof read by a native speaker.

Answer:

Thank the reviewer for the valuable comment.

We are very sorry for our misleading description. In the experiment, only a broad spontaneous emission spectrum is observed when the power density is lower than 23 μJ/cm2 per pulse. Hence, several discrete narrow peaks are clearly observed when the power density exceeds 23 μJ/cm2 per pulse, indicating that the coherent resonant feedback is built up in the polymer film. The light output and light input curve (L-L curve) exhibits a typical“S” shape, a clear indication of the transition from a spontaneous emission to an amplified spontaneous emission to stimulated emission with increasing power densities. This significant feature indicates that the coherent random laser has a working threshold about 23 μJ/cm2 per pulse. We have replaced the sentence “where a solid blue line changes abruptly” with “the coherent random laser has a working threshold of 23 μJ/cm2 per pulse” in the revision manuscript. The corresponding explanation and the figure are added in the revised manuscript. [Lines 133 on the page 4]

  1. The authors state that the effective cavity length amounts to 36.14 micrometers based on equation 1 and the Fourier component p1, which amounts to 17.14 micrometers as shown in figure 2) (d). It is absolutely unclear to me how the authors derive this value from figure 2) (d). The signal is certainly not periodic with a period of 17.14 micrometers as there are no peaks at, e.g., 51.42 micrometers or 85.7 micrometers. Are only even harmonics expected or what is the reason for this?

Answer:

Thank the reviewer for the valuable comment.

To calculate the effective cavity length of the coherent random laser, the power Fourier transforms (PFT) of the spectrum are calculated.

The spatial dimensions can be calculated by

                                                              (1)

where pm is a Fourier component, m is the order of the Fourier harmonics, n is the refraction index of the gain medium, and Lc is the localized cavity dimension. Most of the time, we will chose the first Fourier component as numerical calculation so that it is closer to the actual situation. In our experiment, we chose the m and n are 1 and 1.49 respectively, and p1 is the 17.14 μm. So, the effective optical cavity length Lc can be calculated as 36.14 μm by Eq.(1). The corresponding explanation and the figure are added in the revised manuscript. [Lines 149 on the page 5]

  1. It is not clear what the term "crook motion" actually refers to in line 130. This might be a non-ideal translation.

Answer:

Thank the reviewer for the valuable comment.

We are very sorry for our misleading description. In the manuscript, the word “crook motion” is only used to describe the moving progress of the translation stage when the polymer film is bent by two translation stages. We have replaced the word “crook motion” with “bending motion” in the revision manuscript. [Lines 173 on the page 6]

  1. In figure 3) (b) it is hard to see all details. Is it really necessary to show the full spectrum down to 570 nm? There is almost no emission between 570 nm and 585 nm.

Answer:

Thank the reviewer for the valuable comment.

The dispersed Au NRs can not only serve as scattering particles to provide coherent feedback on the large-area polymer film, but also enhance the PL emission due to SPR. Fig. R4b presents the schematic diagram of the principle of bending strain. Additionally, the center wavelength of the coherent random laser is blue-shifting due to the decrease of the distances between Au NRs when the polymer film is bending stain as shown Fig. R4b, which altered the plasmon interaction and the scattering. Here, the length (L=1 cm) is defined as the original length of the polymer film without exerting any bending strain and the ΔL is the bending length when the polymer film is under bending strain.

We have changed the spectrum ranging from 600 nm to 650 nm (in Fig. R4c-d). The evolution of the PL spectra of the polymer film ornament with Au NRs under different bending strains from 0 mm to 5 mm and recoverable, which exhibited good repeatability as shown Fig. R4c-d. In our experiment, the blue-shifting is over 11 nm (center wavelength ranges from 624 nm to 612.6 nm) with a bending strain of 5 mm as shown in Fig. R4e. The experimental data are the average wavelength of Fig. R4c. The corresponding explanation and the figures are added in the revised manuscript.

Figure R4. The optical spectrum characteristics are under the bending strain. (a) A schematic diagram of the experimental device. (b) The schematic diagram of the principle. The original length (L) is 1 cm, ΔL is the length variation when the polymer film is under bending strain. (c-d) The signals of the coherent random laser are detected under different bending strains. Insets exhibit schematic diagrams under bending strain. (e) The lasing modes are a function of the degree of bending. The experimental data are the center wavelength of (c). Top right inset: the two states of a polymer film transfer to the wrist. [Lines 190 on the page 6]

  1. At the end of the caption of figure 3, the authors refer to a "top left inset", which is, however, located in the top right corner.

Answer:

Thank the reviewer for the valuable comment.

We are very sorry for our misleading description. We have replaced the word “top left inset” with “top right inset” in the revision manuscript. [Lines 197 on the page 7]

  1. I have no idea how the authors arrive at the data shown in figure 3(e). They show that the lasing mode shifts from about 631.5 nm with no strain applied to roughly 619.5 nm for the strongest strain and state that this data is taken from 3(c). Looking at the spectrum in 3(c) when no strain is applied, one can clearly see that there is either no or very little emission at 631.5 nm. Accordingly, the authors should describe very carefully how they extract the lasing wavelength shown in 3(e) from the data shown in figure 3(c). Currently, the data shown in 3(e) does not seem reliable. The authors also do not state which pump fluence has been used to produce these results. As it is not directly clear whether the spectra shown in 3(c) and in 2(a) look similar at all, this might be valuable information.

Answer:

Thank the reviewer for the valuable comment.

We have replaced the expression “peak wavelength” with “average wavelength with error bars in the dependency graphs” as shown in Fig. R5e.

The emission spectra is not collected on the same day. The pump region of polymer film is different, and the pump position is also different from the same time. There are some reasons for the different emission spectra, the excitation power and the surrounding conditions might affect the emission spectra. There are some competition between different laser modes, so the emission spectra is not similar at all.

In the experiment, the polymer film is bent by two translation stages to imitate the human’s bending strain as illustrated in Fig. R5a, the schematic diagram of the principle of bending strain as shown in Fig. R5b. Here, the length (L=1 cm) is defined as the original length of the polymer film without exerting any bending strain and the ΔL is the bending length when the polymer film is under bending strain. Fig. R5c-d show the evolution of the PL spectra of the polymer film ornament with Au NRs under different bending strains from 0 mm to 5 mm and recoverable, which exhibited good repeatability. The pump fluence is about 80.26 μJ/cm2 per pulse. Additionally, the wavelength of the random lasing is blue-shifting due to the decrease of the distances between Au NRs, which altered the plasmon interaction and the scattering. In our experiment, the blue-shifting is over 11 nm (average wavelength ranges from 624 nm to 612.6 nm) with a bending strain of 5 mm as shown in Fig. R5e. The experimental data are the average wavelength of Fig. R5c.

Figure R5. The optical spectrum characteristics are under the bending strain. (a) A schematic diagram of the experimental device. (b) The schematic diagram of the principle. The original length (L) of the polymer film is 1 cm, ΔL is the bending length when the polymer film is under bending strain. (c-d) The signals of the coherent random laser are detected under different bending strains. Insets exhibit schematic diagrams under bending strain. (e) The lasing modes are function of the degree of bending. The blue-shifting is over 11 nm. The experimental data are the average wavelength of (c). Top right inset: the two states of a polymer film transfer to the wrist. [Lines 190 on the page 6]

  1. When looking at figures 4(c) and 4(d), the relaive humidities are inversely color coded in both figures - the color used for 40% RH in panel (c) is used for 90% RH in panel (d), the color used for 48% RH in panel (c) is used for 76% RH in panel (d) and so on. This is very confusing. It would be very helpful for the reader if the authors used a consistent color for the different RH values.

Answer:

Thank the reviewer for the valuable comment.

We are very sorry for our misleading description. We have replaced the expression “peak wavelength” with “average wavelength with error bars in the dependency graphs” as shown in Fig. R6d. We have used a consistent color for the different RH values as shown in Fig. R6c-d. The corresponding explanation and the figure are added in the revised manuscript.

Figure R6. Measurement of small motion signals. (a) The wearable sensors for on-skin applications. (b) The schematic diagram of the principle of RH sensing. (c) The emission spectra of random laser with the RH ranging from 40% to 88%. (d) Plots describing emission peaks as a function of relative humidity. [Lines 219 on the page 7]

  1. The inset of figure 2(c) might benefit from some indication of what the spatial size of the region shown actually is.

Answer:

Thank the reviewer for the valuable comment.

We are very sorry for our misleading description. The inset is only used to illustrate the diameter of pump spot, which is about 0.28 cm. The corresponding explanation and the figure are added in the revised manuscript. [Lines 155 on the page 5]

  1. In figure 2(b), the y-axes are not very meaningful. For example, the plot showing the FWHM is more or less meaningless as pretty much any change in the FWHM will not be visible on a scale going up to a FWHM of 10 nm. Maybe it would be helpful to divide panel (b) into three small horizontal subpanels with meaningful ranges of each y-axis.

Answer:

Thank the reviewer for the valuable comment.

Fig. R7 presents the output lasing intensity and full width at half maximum (FWHM) of the large-area polymer film through pumping at different positions with a distance step of 1 cm. The FWHM of coherent random lasing has almost no change, which is about 0.4 nm, indicating that the large-area coherent random lasing has excellent optical stability. The position of the optical excitation source starts from zero onward with an increment of 1 cm covering a total distance of 10 cm as shown the inset in Fig. R7. The corresponding explanation and the figure are added in the revised manuscript.

Figure R7. The FWHM (red squares) and the output intensity (blue squares) are at different pump positions on the large-area polymer film. Inset: the distance between two pump positions is about 1 cm. [Lines 155 on the page 5]

  1. The intensity increase shown in figure 2(c) shows a completely linear slope above threshold, which is unusual for a lasing transition. It is not clear how the authors deduce this set of data from 2(a). Is this the whole intensity integrated over all possible emission modes? If so, the authors should make that clear. Also, the right y-scale in figure 2(c) is far from ideal. The reader cannot judge whether the FWHM of the lasing mode changes at all above the threshold as the range is so large. An inset or a zoomed-in version of the FWHM values above the threshold might help the reader to understand what is going on.

Answer:

Thank the reviewer for the valuable comment.

This is the whole intensity integrated over all possible emission modes. The maximum intensity of random laser is chosen in Fig. R8c. Inset on the right is the FWHM values above the threshold. So we can judge the FWHM of the lasing mode changes at all above the threshold. The FWHM of lasing mode is about 0.4 nm. In the experiment, only a broad spontaneous emission spectrum is observed when the power density is lower than 23 μJ/cm2 per pulse. Hence, several discrete narrow peaks are clearly observed when the power density exceeds 23 μJ/cm2 per pulse, indicating that the coherent resonant feedback is built up in the polymer film (in Fig. R8a).

Fig. R8b presents the output intensity and FWHM of the large-area polymer film through pumping at different positions with a distance step of 1 cm. The FWHM of coherent random lasing has almost no change, which is about 0.4 nm, indicating that the large-area coherent random lasing has excellent optical stability. The position of the optical excitation source starts from zero onward with an increment of 1 cm covering a total distance of 10 cm as shown the inset of Fig. R8b.

Fig. R8c presents evolution of the random lasing intensity (blue circles) and FWHM (red triangles) as a function of the pump fluences. The light output and light input curve (L-L curve) exhibits a typical“S” shape, a clear indication of the transition from a spontaneous emission to an amplified spontaneous emission to stimulated emission with increasing power densities (in Fig. R8c). This significant feature indicates that the coherent random laser has a working threshold of about 23 μJ/cm2 per pulse. The left illustration indicates the optical image under the pumping. The right inset shows partial enlarged of pump fluence and FWHM. The diameter of pump spot is about 0.28 cm.

To calculate the effective cavity length of the coherent random laser, the PFT of the spectrum are calculated. The effective optical cavity length Lc can be calculated as 36.14 μm by Eq.(1) as shown in Fig. R8d. The corresponding explanation and the figure are added in the revised manuscript.

Figure R8. Spectra characterization of the coherent random laser. (a) The evolution of the random lasing under different pumping fluences ranges from 10 μJ/cm2 per pulse to 90 μJ/cm2 per pulse. Inset: the enlarged view of the emission spectrum with 23 μJ/cm2 per pulse. (b) The FWHM (red squares) and the output intensity (blue squares) are at different pump positions. (c) The pump power and FWHM are at different power densities and the lasing threshold of 23 μJ/cm2 per pulse. Scale bar =0.3 cm. (d). The power Fourier transform of the random laser spectra, the effective optical cavity length Lc is calculated to be 36.14 μm. [Lines 155 on the page 5]

We really appreciate that you read our manuscript carefully and give these valuable suggestions.

Special thanks to you for your valuable comments.

References

  • Z.Y.; Zhai. T.R.; Shi. X.Y.; Tong. J.H.; Wang. X.L.; Deng. J.X. Multifunctional sensing based on an ultrathin transferrable microring laser. ACS Appl. Mater. Interfaces 2021, 13, 19324-19331, doi: 10.1021/acsami.1c03123.
  • Desimoni, E.; Brunetti, B. About estimating the limit of detection by the signal to noise approach. Pharm. Anal. Acta 2015, 6, 1000355, doi: 10.4172/2153-2435.1000355.
  • Y.Q.; Liu. Y.X.; Zhong. D.L.; Nikzad. S.; 1 , Liu. S.H.; Yu. Z.; Liu. D.Y.; Wu. H.C.; Zhu. C.X.; Li. J.X.; Tran. H.; Tok. J.B.; Bao. Z.N. Monolithic optical microlithography of high-density elastic circuits. Science 2021, 373, 88-94, doi: 10.1126/science.abh3551.

Round 2

Reviewer 2 Report

The authors have resubmitted a revised version of their manuscript on wearable random lasers. They incorporated a quite long list of changes which indeed improved the manuscript. Also many language problems and non-ideal phrasings have been removed by the authors. At some points, the language might still be polished, but this can be done at the copyediting stage.

In terms of the results shown, the presentation of the results has improved and it is now easier for the reader to follow the conclusions presented by the authors. Especially the authors' response to the questions raised about figure 3 are very helpful. Still, I would suggest to change the label of the y-axis in figure 3(e) to "average wavelength" to make it more unambiguous and to explicitly mention that not all spectra were recorded at the same time and the same position, so the reader can understand why the spectra look different.

Besides that, it is my opinion that the present manuscript can be published following the usual "polishing" procedures of the copyediting phase.

Author Response

Thank the reviewer for the valuable comment.

We are very sorry for our inappropriate description. In the manuscript, the word “wavelength” is only used to describe the evolution of the average wavelength of the random lasing under the bending strain. We have replaced the label of the y-axis to “average wavelength” in the revision manuscript as shown in Fig. R1.

The spectra are different from others due to the spectra were recorded at the different time and the different position. The corresponding explanation and the figures are added in the revised manuscript.

Figure R1. The optical spectrum characteristics are under the bending strain. The blue-shifting of the average wavelength are a function of the degree of bending. [Lines 186, 188 on the page 6; Lines 202 on the page 7]

We really appreciate that you read our manuscript carefully and give these valuable suggestions.

Special thanks to you for your valuable comments.
